# Single and Competitive Adsorption Behaviors of Cu$^{2+}$, Pb$^{2+}$ and Zn$^{2+}$ on the Biochar and Magnetic Biochar of Pomelo Peel in Aqueous Solution

Qianlan Wu [1], Shuzhen Dong [2], Lijun Wang [1,*] and Xiaoyun Li [1]

[1] Department of Environmental Science and Engineering, School of Geography and Tourism, Shaanxi Normal University, Xi'an 710119, China; wuqianlan97@163.com (Q.W.); lixiaoyun518@163.com (X.L.)

[2] Shangluo City Meteorological Bureau, Shangluo 726000, China; dsz181994@163.com

\* Correspondence: wanglijun@snnu.edu.cn

**Abstract:** As an environment-friendly material, biochar has been used to remove heavy metals from wastewater, and the development of cost-effective biochar has been an emerging trend. However, limited studies consider the competitive adsorption of co-existing metals and the separation efficiency of absorbent and solution after adsorption. In this study, pomelo peel was used to prepare biochar (BC) and magnetic biochar (MBC) at different temperatures. Then, the physicochemical properties of the biochars were characterized and the adsorption characteristics of Cu$^{2+}$, Pb$^{2+}$, and Zn$^{2+}$ on the biochars in single, binary, and ternary metal systems were investigated. The results showed that both pyrolysis temperature and magnetization could affect the adsorption capacity of biochar. The adsorption kinetic and thermodynamic processes could be well described by the pseudo-second-order kinetic model and Langmuir model. The adsorption isotherm types of Pb$^{2+}$ and Zn$^{2+}$ changed in the binary metal condition. The competitive adsorption order of three heavy metal ions in ternary metal adsorption was Pb$^{2+}$ > Cu$^{2+}$ > Zn$^{2+}$. The MBC of 500 °C showed a good adsorption capacity to Pb$^{2+}$ in the co-existing environment, and the maximum adsorption capacity was 48.74 mmol g$^{-1}$. This study also provided technical support for the utilization of pomelo peel and the engineering application of biochar.

**Keywords:** biochar; magnetic biochar; bivalent metal; adsorption; possible mechanism

## 1. Introduction

Most parts of China are facing a huge challenge from water pollution due to the developing manufacture of chemical, coal, electronic, and agricultural products [1,2]. Some rivers, lakes, and sediments in China have been exposed to moderate ecological risk from heavy metals, with even potential adverse biotoxic effects [3–6]. Heavy metal pollution in water has attracted much attention due to its persistence, bio-accumulation, non-degradation, and toxicity. Furthermore, heavy metals can circulate and accumulate in organisms through the food chain, posing a great potential threat to organisms, especially for Cu, Pb, and Zn [7,8]. Currently, the removal efficiency of heavy metals varies in treatment, and most of the methods are not practical due to the production of secondary pollution, high-cost operation, and maintenance, etc. [9]. Thus, it is necessary to seek appropriate treatment methods for heavy metals. Advancements in adsorption offer a better method for heavy metal removal, while a variety of heavy metal ions generally co-exist in the actual environment [10–12]; therefore, the competitive adsorption of co-existing metal ions in water should be considered in practical treatment.

Biochar is generally produced by the pyrolysis of biomass (e.g., wood, poultry litter, and crop residues) under a limited supply of oxygen at relatively low temperatures (<700 °C) [13,14]. Biochar has been widely applied in water treatment as a low-cost adsorbent due to its large surface area, porous structure, abundant surface functional groups,

and high cation exchange capacity [15,16]. Usually, the adsorption capacity of unmodified biochar is relatively small, and it is also not easy to separate from solution due to its low density. Thus, studies have tried various modifications, including chemical, physical, mineral impregnation, and magnetic modifications, aiming to obtain efficient biochar [17]. Among these, magnetic modifications may improve the separation efficiency of biochar from water, which makes biochar more practical in engineering application. Yin et al. [18] prepared magnetic biochar showing a good adsorption capability for Cr(VI) from aqueous solution. Kim et al. [19] reported that Fe-modified biochar was an efficient adsorbent for As(III). In addition, a previous study indicated that a magnetic field improves the filtration process without increasing the number of compounds introduced into the water treatment system or the environment [20]. Iwona Skoczko et al. [21] pointed out that filtration with a magnetic field on activated alumina for the removal of selected elements is more effective than filtration without a magnetic field. Therefore, choosing green feedstocks and making proper magnetic modifications are critical for the development of efficient biochar adsorbents.

Pomelo peel is a kind of common agricultural waste which has natural adsorption properties. Wang et al. [22] reported that $FeCl_3$-modified pomelo peel could be used for the removal of Cr(VI); Low and Tan [23] demonstrated that ultrasound pre-treatment of pomelo peel enhanced its ability to remove methylene blue from aqueous solution. However, these studies did not take compound pollution into account. Furthermore, pomelo peel contains a great deal of lignin, cellulose, and pectin, which can form pore structures on the surface of biochar after carbonization. Hence, this study attempted to carbonize pomelo peel into biochar and then magnetize the biochar, aiming to obtain adsorbents with excellent physiochemical properties and easy separation.

Herein, the biochar (BC) and magnetic biochar (MBC) of pomelo peel were employed to demonstrate the absorption behaviors and mechanisms of $Cu^{2+}$, $Pb^{2+}$, and $Zn^{2+}$ in single, binary, and ternary metal systems. The specific objectives of this study were to (1) reveal the adsorption characteristics and competitive mechanisms of single, binary, and ternary metal ions on BCs and MBCs by batch experiments and (2) provide technical support for both the pollution control of heavy metals and engineering application of pomelo peel biochar.

## 2. Materials and Methods

### 2.1. Materials and Reagents

Pomelo peel was obtained from a fruit market in Shaanxi Normal University (34.21° N, 108.96° E, Xi'an, China). All chemicals and reagents utilized in the experiments were of analytical grade. $FeCl_3 \cdot 6H_2O$, $FeCl_2 \cdot 4H_2O$, NaOH, KBr, HCl, NaOH, and $Pb_2(NO_3)_3$ were bought from Sinopharm Chemical Reagent Co., Ltd (Shanghai, China). $CuCl_2 \cdot 2H_2O$ and $ZnCl_2$ were purchased from Kemiou, Tianjin Chemical Reagent Co., Ltd (Tianjin, China).

### 2.2. Preparation of Biochars

The preparation process of biochars is given in Figure S1. In details, the pomelo peel was first washed using distilled water, air-dried, and then dried in an oven overnight at 60 °C. The dried pomelo peel was comminuted, passed through a 100-mesh sieve, and then carbonized for 2 h in a muffle furnace at temperatures of 400, 500, and 600 °C within nitrogen atmosphere. The pomelo peel biochars prepared at different temperatures were washed to neutral with distilled water, dried at 70 °C, and labeled as BC400, BC500, and BC600, respectively. The magnetic biochar was prepared using the chemical co-precipitation method with the previously prepared pomelo peel biochar as material [24,25]. To be specific, 4 g of a certain prepared pomelo peel biochar was added in a three-neck flask (evacuated with nitrogen in advance) with a 400 mL solution containing 6.4870 g $FeCl_3 \cdot 6H_2O$ and 2.3857 g $FeCl_2 \cdot 4H_2O$ (the molar ratio of $Fe^{3+}$: $Fe^{2+}$ was 2:1), and they were stirred for 2 h in nitrogen environment. After 100 mL NaOH solution (0.1 mol $L^{-1}$) was added drop by drop (one drop per second), they were aged for 1 h in a 70 °C water bath. Then, they were separated by a magnetic field, and the obtained magnetic biochars were

also washed with distilled water to neutral and dried at 70 °C. The magnetic biochars were ground and labeled as MBC400, MBC500, and MBC600, respectively.

*2.3. Characterization of Biochars*

The specific surface area of the biochars was measured with a surface area and porosity analyzer (ASAP 2460, Micromeritics, Atlanta, GA, USA). Before testing, 200 mg of biochar was weighed and dried. The Brunauer–Emmett–Teller (BET) equation was used to calculate the specific surface area of biochar after vacuum degassing at 450 °C for 6 h. The pore size distribution was determined by the adsorption and desorption isotherm of $N_2$ and calculated using the Barrett–Joyner–Halenda (BJH) method. The pore volume was calculated by the adsorption capacity at relative pressure of $P/P_0 = 0.20$. The analytical samples were first prepared using the KBr pressed-disk technique: 1 mg of sample dried at 70 °C for 24 h and 50 mg of KBr were ground in a quartz mortar and pressed as a thin slice. Then, the Fourier-transform infrared (FTIR) spectra of the biochars before and after adsorption were obtained using FTIR spectroscopy (TENSOR27, Bruker, Ettlingen, Germany) from 400 to 4000 cm$^{-1}$ with a resolution of 4 cm$^{-1}$. X-ray diffraction (XRD) analysis of the biochars before and after adsorption was conducted with an X-ray diffractometer (D8 Advance, Bruker, Germany) at 40 kV and 40 mA (Cu K$\alpha$ radiation). The data scanning step was 0.02°, and the scanning range (2θ) was 5–80°.

*2.4. Adsorption Experiments*

The adsorption of three heavy metal ions (i.e., $Cu^{2+}$, $Pb^{2+}$, and $Zn^{2+}$) on the biochars was studied. In the adsorption kinetics studies, 0.05 g of the biochars was weighed in a 20 mL headspace vial and mixed with 10 mL solution of 40 mg L$^{-1}$ $Cu^{2+}$ (the concentration of $Pb^{2+}$ and $Zn^{2+}$ was 100 and 20 mg L$^{-1}$, respectively), and then, the mixture was shaken for 10, 20, 30, 45, 60, 120, and 180 min in a constant-temperature water bath vibrator (180 rpm, 25 ± 1 °C). Adsorption isotherms of single adsorbates were obtained by adding 0.05 g of the biochars into corresponding 20 mL headspace vials; then, 10 mL solution of different concentrations of $Cu^{2+}$ (10, 20, 40, 50, 60, and 80 mg L$^{-1}$) was added. The concentration gradient of $Pb^{2+}$ was 50, 75, 100, 120, 150, and 200 mg L$^{-1}$, and the gradient for $Zn^{2+}$ was 5, 10, 20, 30, 40, and 50 mg L$^{-1}$. They were shaken for 1 h in a constant-temperature water bath vibrator (180 rpm; 25 ± 1 °C, 35 ± 1 °C and 45 ± 1 °C).

BC500 and MBC500 were selected as the adsorbents of competitive adsorption. For the binary systems, the concentration of the target metal ion was adjusted based on the single-metal adsorption and the other two metal ions were respectively taken as the competitive ions with constant concentrations. There were 6 systems in this study: $Zn^{2+}$ (20 mg L$^{-1}$)—$Cu^{2+}$ (10, 20, 40, 60, and 80 mg L$^{-1}$); $Pb^{2+}$ (60 mg L$^{-1}$)—$Cu^{2+}$ (10, 20, 40, 60, and 80 mg L$^{-1}$); $Zn^{2+}$ (20 mg L$^{-1}$)—$Pb^{2+}$ (20, 40, 60, 80, and 100 mg L$^{-1}$); $Cu^{2+}$ (40 mg L$^{-1}$)—$Pb^{2+}$ (20, 40, 60, 80, and 100 mg L$^{-1}$); $Cu^{2+}$ (40 mg L$^{-1}$)—$Zn^{2+}$ (10, 20, 40, 60, and 80 mg L$^{-1}$); and $Pb^{2+}$ (60 mg L$^{-1}$)—$Zn^{2+}$ (10, 20, 40, 60, and 80 mg L$^{-1}$). Each solution was mixed with 0.05 g certain biochar and then shaken for 1 h in a constant-temperature water bath vibrator (180 rpm, 25 ± 1 °C).

For ternary metal competitive adsorptions, the concentration of each metal ion was the same in the mixed solution and the concentration gradient of all the studied metal ions was 10, 20, 40, 60, 80, and 100 mg L$^{-1}$. Each gradient was mixed with 0.05 g of a certain biochar and then shaken for 1 h in a constant-temperature water bath vibrator (180 rpm, 25 ± 1 °C).

All adsorption experiments were repeated three times. The concentration of the studied heavy metal ions was determined using an inductively coupled plasma optical emission spectrometer (Acros, Spetro, Kleve, Germany).

*2.5. Data Analysis*

The adsorption capacity at time t ($Q_t$), equilibrium adsorption capacity ($Q_e$), and removal efficiency (E) of heavy metal ions were calculated as follows:

$$Q_t = (C_0 - C_t) \times V/M \tag{1}$$

$$Q_e = (C_0 - C_e) \times V/M \tag{2}$$

$$E(\%) = (C_0 - C_e)/C_0 \times 100 \tag{3}$$

where $C_0$, $C_t$, and $C_e$ (mmol L$^{-1}$) are the initial, t time, and equilibrium concentrations of heavy metal ions, respectively; M (g) is the amount of biochar; and V (mL) is the volume of heavy metal solution.

The adsorption kinetics results were fitted by a pseudo-first-order model, pseudo-second-order kinetics model, and intraparticle diffusion equation, which were expressed as follows [26]:

$$\ln(Q_e - Q_t) = \ln Q_e - k_1 t \tag{4}$$

$$t/Q_t = 1/k_2 Q_e{}^2 + t/Q_e \tag{5}$$

$$Q_t = K_i t^{0.5} + C \tag{6}$$

where $k_1$ (min$^{-1}$) and $k_2$ (g mmol$^{-1}$ min$^{-1}$) are the rate constants for pseudo-first-order and pseudo-second-order adsorption, respectively; $K_i$ is the intraparticle diffusion rate constant (mmol g$^{-1}$ min 0.5), and C is a constant providing the thickness of the boundary layer.

The adsorption thermodynamic results were fitted using the Langmuir and Freundlich models, which were expressed as follows [27]:

$$C_e/Q_e = 1/(K_L \times Q_m) + C_e/Q_m \tag{7}$$

$$\ln Q_e = \ln K_f + 1/n \times \ln C_e \tag{8}$$

where $Q_m$ (mmol g$^{-1}$) is the maximum adsorption capacity, $Q_e$ (mmol g$^{-1}$) is the equilibrium adsorption capacity, $K_L$ (L mmol$^{-1}$) is the Langmuir constant, and $K_f$ [(mmol g$^{-1}$) (mmol L$^{-1}$)$^{1/n}$] and $1/n$ are the Freundlich constants.

Adsorption thermodynamic parameters, i.e., Gibbs free energy ($\Delta G^0$, kJ mol$^{-1}$), enthalpy of adsorption ($\Delta H^0$, kJ mol$^{-1}$), and entropy of adsorption ($\Delta S^0$, J mol$^{-1}$ K$^{-1}$), were calculated as follows [28]:

$$K_c = Q_e/C_e \tag{9}$$

$$\Delta G^0 = -RT \ln K_c \tag{10}$$

$$\ln K_c = \Delta S^0/R - \Delta H^0/RT \tag{11}$$

where $K_C$ is the equilibrium constant, R is the gas constant (8.3145 J mol$^{-1}$ K$^{-1}$), and T is the absolute temperature at 298 K.

Microsoft Office Excel (2010) and SPSS 20 (SPSS Inc., Chicago, IL, USA) were used for data processes. The experimental data were fitted using Origin 9.0 (OriginLab, Northampton, MA, USA).

## 3. Results and Discussion

### 3.1. Single-Metal Adsorption onto Biochars

3.1.1. Adsorption Kinetics

As shown in Figure S2, the adsorption of Pb$^{2+}$ on BCs and MBCs and the adsorption of Zn$^{2+}$ on BCs showed two distinct phases: a rapid phase over the first 60 min and a second slow phase, until reaching equilibrium at approximately 120 min. The adsorption of Zn$^{2+}$ on MBCs could reach equilibrium at around 60 min. For the adsorption of Cu$^{2+}$, the adsorption capacity increased with the contact time, and there was no obvious plateau for BCs and MBC600. The possible reasons for the two different adsorption phases are that there are a large number of absorbable sites at the initial stage, and then, the repulsion between solute molecules in solid and bulk phases prevents further adsorption after a period of time.

In this study, the pseudo-second-order model was more suitable for the adsorption kinetic processes of metal ions on BCs and MBCs than the pseudo-first-order model; therefore, only the fitting results of experimental data by the pseudo-second-order model

are given in Table S1. As presented in Table S1, the fitting correlation coefficients ($R^2$) of the pseudo-second-order model were greater than 0.9930, and the adsorption capacity values ($Q_{e,cal}$) acquired from the pseudo-second-order model were also close to the experimental values ($Q_{e,exp}$). The well-fitting results indicated that the pseudo-second-order kinetic model could be used to describe the adsorption kinetics processes of the studied metal ions onto the BCs and MBCs, and the adsorption process of three kinds of metal ions depended on the rate-limiting chemisorption [29,30]. Furthermore, the $Q_{e,exp}$ and $Q_{e,cal}$ of the studied metal ions increased with the increasing pyrolysis temperature, which may contribute to the larger surface area and total pore volume of biochars prepared at a high temperature. These structures could provide more adsorption site for metal ions.

The intraparticle diffusion model of Weber–Morris has often been used to determine whether intraparticle diffusion is a rate-limiting step. In this model, the plot of $Q_t$ versus $t^{0.5}$ should be linear if intraparticle diffusion is involved in the adsorption process, and intraparticle diffusion is the only rate-limiting step if the plot passes through the origin. It has also been reported that there are two or more steps governing the adsorption process when $Q_t$ versus $t^{0.5}$ is multi-linear [31]. As shown in Figure S3, the fitting curves exhibited multi-linear relationships, indicating that two or more adsorption stages occurred. The large slopes in the first stage represented surface or film diffusion; the second linear sections represented a gradual adsorption stage where intraparticle diffusion was a rate-limiting step [32]. In addition, the curves did not pass through the origin, indicating that intraparticle diffusion was not the only rate-limiting step. As seen in Table S2, $K_{i1}$ was larger than $K_{i2}$, suggesting that the adsorption rate of liquid membrane diffusion in the first stage was higher than that of particle diffusion in the second stage. The C value in the model providing the information related to the thickness of boundary layer suggested that surface diffusion played an important role in the rate-limiting step based on the obtained larger intercepts (Table S2) [26].

### 3.1.2. Adsorption Isotherms

The Langmuir and Freundlich adsorption isotherm models were applied to evaluate the adsorption thermodynamic processes of the studied metal ions on BCs and MBCs. The experimental and fitting results obtained at 25, 35, and 45 °C are shown in Figures S4–S6 and Tables S3–S5. For the adsorption of $Cu^{2+}$ on the BCs, the Langmuir model had high correlation coefficients ($R^2 > 0.9370$) relative to the Freundlich model (Figure S4, Table S3), suggesting that the adsorption process of $Cu^{2+}$ on BCs was a monolayer adsorption process. The fitting degree of $Cu^{2+}$ adsorption on the MBCs was lower than that on the BCs, while the $R^2$ of the Langmuir model was still larger than that of the Freundlich model. According to the $Q_m$ obtained from the Langmuir model, the MBCs had a larger adsorption capacity than the BCs, and the maximum adsorption capacity of $Cu^{2+}$ was 140.44 mmol $g^{-1}$ (MBC500, 45 °C). For the adsorption of $Pb^{2+}$, the fitting degree of the Langmuir model was significantly higher than that of the Freundlich model (Figure S5, Table S4). At the same contact temperature, the $Q_m$ obtained from the Langmuir model for MBCs was higher than that for BCs. The maximum adsorption capacity of $Pb^{2+}$ was 155.35 mmol $g^{-1}$ (MBC500, 45 °C). The adsorption of $Zn^{2+}$ had different characteristics compared with $Cu^{2+}$ and $Pb^{2+}$, and the correlation coefficients of the Langmuir and Freundlich models were relatively close (Figure S6, Table S5). Meanwhile, BC600 showed the best adsorption capacity for $Zn^{2+}$ (86.44 mmol $g^{-1}$) at 45 °C. In conclusion, the adsorption capacity of $Pb^{2+}$ was close to $Cu^{2+}$, and $Zn^{2+}$ was the smallest. This result was consistent with the study result of Park et al. [33]. In addition, the adsorption capacities of BCs and MBCs to the tested metal ions increased with the increasing temperature, demonstrating that the adsorption of metal ions on both BCs and MBCs was an endothermic process.

### 3.1.3. Analysis of Thermodynamic Parameters

The thermodynamic parameters including the $\Delta G^0$, $\Delta H^0$, and $\Delta S^0$ of adsorption are listed in Table 1. The negative values of $\Delta G^0$ in the studied temperature range of 25–45 °C demonstrated that the adsorption of $Cu^{2+}$, $Pb^{2+}$, and $Zn^{2+}$ on BCs and MBCs was feasible and spontaneous in nature [34]. In addition, the absolute values of $\Delta G^0$ increased with the increasing test temperature. This result showed that the adsorption was a chemisorption process, and the increasing reaction temperature increased the number of active molecules, which was favorable to the spontaneous reaction [35]. It was consistent with the above analysis that the maximum adsorption capacity increased with the increasing adsorption reaction temperature. The positive values of $\Delta H^0$ illustrated that the adsorption was an endothermic process. All values of $\Delta S^0$ were positive, illustrating that the solid/solution interface of metal ions and BCs/MBCs was disordered [36].

**Table 1.** Adsorption thermodynamic parameters for $Cu^{2+}$, $Pb^{2+}$, and $Zn^{2+}$ on biochars (BCs) and metallic biochars (MBCs).

| Metal Ions | Biochar | $\Delta G^0$ (kJ mol$^{-1}$) | | | $\Delta H^0$ (kJ mol$^{-1}$) | $\Delta S^0$ (J mol$^{-1}$ K$^{-1}$) | $R^2$ |
|---|---|---|---|---|---|---|---|
| | | 25 °C | 35 °C | 45 °C | | | |
| $Cu^{2+}$ | BC400 | −13.00 | −13.44 | −13.87 | 19.70 | 43.57 | 0.9934 |
| | BC500 | −9.07 | −9.37 | −9.67 | 14.30 | 30.38 | 0.9787 |
| | BC600 | −17.78 | −18.38 | −18.98 | 27.78 | 59.76 | 0.9974 |
| | MBC400 | −28.56 | −29.51 | −30.47 | 31.09 | 95.72 | 0.9787 |
| | MBC500 | −9.11 | −9.42 | −9.73 | 10.87 | 30.55 | 0.9397 |
| | MBC600 | −7.22 | −7.46 | −7.71 | 7.71 | 20.98 | 0.9233 |
| $Pb^{2+}$ | BC400 | −46.29 | −47.85 | −49.40 | 50.08 | 155.18 | 0.9831 |
| | BC500 | −16.34 | −16.89 | −17.43 | 19.16 | 54.76 | 0.9830 |
| | BC600 | −17.01 | −17.58 | −18.15 | 12.56 | 57.13 | 0.9978 |
| | MBC400 | −12.90 | −13.34 | −13.77 | 13.72 | 43.25 | 0.9810 |
| | MBC500 | −17.81 | −18.40 | −19.00 | 18.04 | 59.69 | 0.9872 |
| | MBC600 | −34.76 | −35.93 | −37.10 | 35.95 | 116.77 | 0.9724 |
| $Zn^{2+}$ | BC400 | −35.02 | −36.20 | −37.37 | 40.77 | 117.39 | 0.9160 |
| | BC500 | −48.86 | −50.49 | −52.13 | 54.71 | 163.76 | 0.9747 |
| | BC600 | −9.99 | −10.32 | −10.66 | 10.38 | 33.54 | 0.9678 |
| | MBC400 | −2.33 | −2.41 | −2.49 | 11.23 | 7.79 | 0.9577 |
| | MBC500 | −2.03 | −2.10 | −2.16 | 9.97 | 6.77 | 0.9110 |
| | MBC600 | −33.46 | −34.58 | −35.70 | 39.71 | 112.40 | 0.9723 |

### 3.2. Binary Metal Adsorption onto Biochars

The results of single-metal adsorption isotherms indicated that compared with BC/MBC600, BC/MBC500 a showed similar or even higher adsorption capacity for the tested mental ions. Taking into account economic costs and low energy consumption, BC500 and MBC500 were selected to study the characteristics of binary and ternary metal co-existing adsorption.

### 3.2.1. Competitive Adsorption of $Cu^{2+}$ with $Zn^{2+}$/$Pb^{2+}$

As shown in Figure 1a,b, when the concentration of $Pb^{2+}$ and $Zn^{2+}$ in the system was fixed, the adsorption capacity of $Cu^{2+}$ on both BC500 and MBC500 first went up and then tended to balance out. Compared with the adsorption capacity of $Cu^{2+}$ in a single-metal system, the adsorption capacity of $Cu^{2+}$ on BC500 and MBC500 in binary metal systems decreased, which was mainly attributed to the competitive adsorption effect of co-existing $Pb^{2+}$ or $Zn^{2+}$. According to the classification of Giles et al. [37], the adsorption isotherms of $Cu^{2+}$ in single and binary metal systems were L-2 type, indicating that the competitive adsorption did not change the adsorption type of $Cu^{2+}$. In addition, the L-2 type adsorption isotherms represented competitive retention, reflecting no surface precipitation.

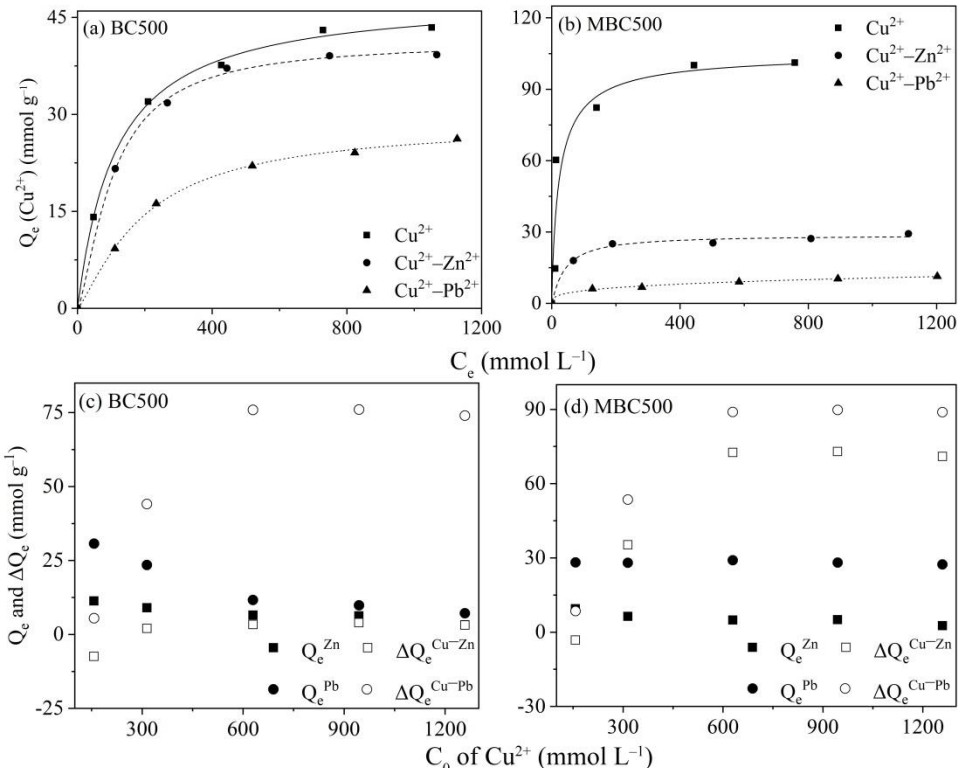

**Figure 1.** Effect of $Zn^{2+}$/$Pb^{2+}$ on adsorption capacity of $Cu^{2+}$ on BC500 (**a,c**) and MBC500 (**b,d**) ($Q_e^{Zn}$ and $Q_e^{Pb}$ represent the adsorption capacities of $Zn^{2+}$ and $Pb^{2+}$ in the binary metal system; $\Delta Q_e^{Cu-Zn}$ and $\Delta Q_e^{Cu-Pb}$ represent the $\Delta Q_e$ of $Cu^{2+}$ in $Cu^{2+}$–$Zn^{2+}$ and $Cu^{2+}$–$Pb^{2+}$ competitive systems).

In order to clearly judge the specific effect of co-existing metal ions on the adsorption of $Cu^{2+}$ on BC500 and MBC500, comparisons between the absolute adsorption inhibition of the target $Cu^{2+}$ ($\Delta Q_e = Q_e - Q_e^{competitor}$) and the adsorption capacity of competitor ($Q_e^{competitor}$ of $Zn^{2+}$/$Pb^{2+}$) were further conducted. As seen from Figure 1c, for BC500, the $Q_e$ of $Pb^{2+}$ decreased with the increasing concentration of $Cu^{2+}$, while the $\Delta Q_e$ increased. This result demonstrated that the inhibition effect of $Pb^{2+}$ on $Cu^{2+}$ was mutual. However, the $Q_e$ of $Zn^{2+}$ kept stable with the increasing concentration of $Cu^{2+}$ and the value of $\Delta Q_e$ was closed to 0, suggesting that there was no interaction between $Zn^{2+}$ and $Cu^{2+}$. As seen in Figure 1d, when the adsorbent was MBC500, the $\Delta Q_e$ of $Cu^{2+}$ was significantly larger than that of BC500 and increased gradually with the increasing $C_0$ of $Cu^{2+}$, indicating that the inhibition effect was amplified on MBC500. Different from BC500, the $Q_e$ of $Pb^{2+}$ remained at a certain level, illustrating that the existence of $Cu^{2+}$ did not affect the adsorption of $Pb^{2+}$ on MBC500. Combined with the fitting results in Table S6, when $Zn^{2+}$ was the adsorption competitive ion, the maximum adsorption capacity (calculated based on the Langmuir model) of $Cu^{2+}$ reduced by 14.4% (BC500) and 72.8% (MBC500); when the competitive ion was $Pb^{2+}$, the adsorption capacity of $Cu^{2+}$ reduced by 40.6% (BC500) and 84.6% (MBC500). These results showed that the inhibition effect of $Pb^{2+}$ on $Cu^{2+}$ was much stronger than that of $Zn^{2+}$, and this inhibition effect was amplified when the adsorbent was MBC500.

### 3.2.2. Competitive Adsorption of $Pb^{2+}$ with $Cu^{2+}$/$Zn^{2+}$

As shown in Figure 2a,b, the adsorption isotherms of $Pb^{2+}$ on BC500 and MBC500 exhibited different shapes in the studied systems and the competition inhibition was obvious. The isotherm of L-1 type of $Pb^{2+}$ in the single-metal system indicated a relatively high affinity of the biochar particles for $Pb^{2+}$. In the binary metal system, the adsorption isotherms of $Pb^{2+}$ changed into L-2 type, suggesting that the competitive ions affected the adsorption of $Pb^{2+}$. Furthermore, the asymptotic plateau of the L-2 curve indicated that a

theoretical monolayer was achieved. As seen in Figure 2c,d, the $Q_e$ of $Cu^{2+}$ and $Zn^{2+}$ and the $\Delta Q_e$ of $Pb^{2+}$ showed the same trend on BC500 and MBC500. Only at a low concentration of $Pb^{2+}$ (96.52 mmol $L^{-1}$), the value of $\Delta Q_e$ was negative, suggesting that the co-existence of $Cu^{2+}/Zn^{2+}$ could promote the adsorption of $Pb^{2+}$. Thereafter, the promotion changed into inhibition and the effect of inhibition increased with the increasing concentration of $Pb^{2+}$. The $Q_e$ of $Cu^{2+}$ and $Zn^{2+}$ on BC500 and MBC500 showed a downward trend, and the adsorption capacities were low, indicating that the existence of $Pb^{2+}$ had a negative effect on the adsorption of $Cu^{2+}$ and $Zn^{2+}$. As shown in Table S7, the adsorption capacity of $Pb^{2+}$ obtained from the Langmuir model decreased by 52.4% (BC500) and 53.7% (MBC500) when competing with $Cu^{2+}$, and by 42.4% (BC500) and 48.2% (MBC500) when competing with $Zn^{2+}$. These results showed that when $Pb^{2+}$ was taken as the target ion, the inhibition effect of $Cu^{2+}$ on $Pb^{2+}$ was slightly stronger than that of $Zn^{2+}$. Combined with the above analysis, there was a strong competitive adsorption when $Pb^{2+}$ and $Cu^{2+}$ co-existed. The possible reason is that BCs and MBCs may have the same adsorption site and the same combination method for $Cu^{2+}$ and $Pb^{2+}$. According to previous studies, the hydration radius of $Pb^{2+}$ (4.01 Å) is smaller than that of $Cu^{2+}$ (4.19 Å) and $Zn^{2+}$ (4.30 Å). The small hydration radius of $Pb^{2+}$ formed smaller three-dimensional hydrated metal clusters or metal hydration shells. The relative weak pore plugging ability enables $Pb^{2+}$ to interact more closely with the biochar. Thus, $Pb^{2+}$ could enter the pores with less mass transfer resistance [38]. In addition, the hydration radii of $Pb^{2+}$ and $Cu^{2+}$ are closer than those of $Pb^{2+}$ and $Zn^{2+}$, so the competition is more obvious.

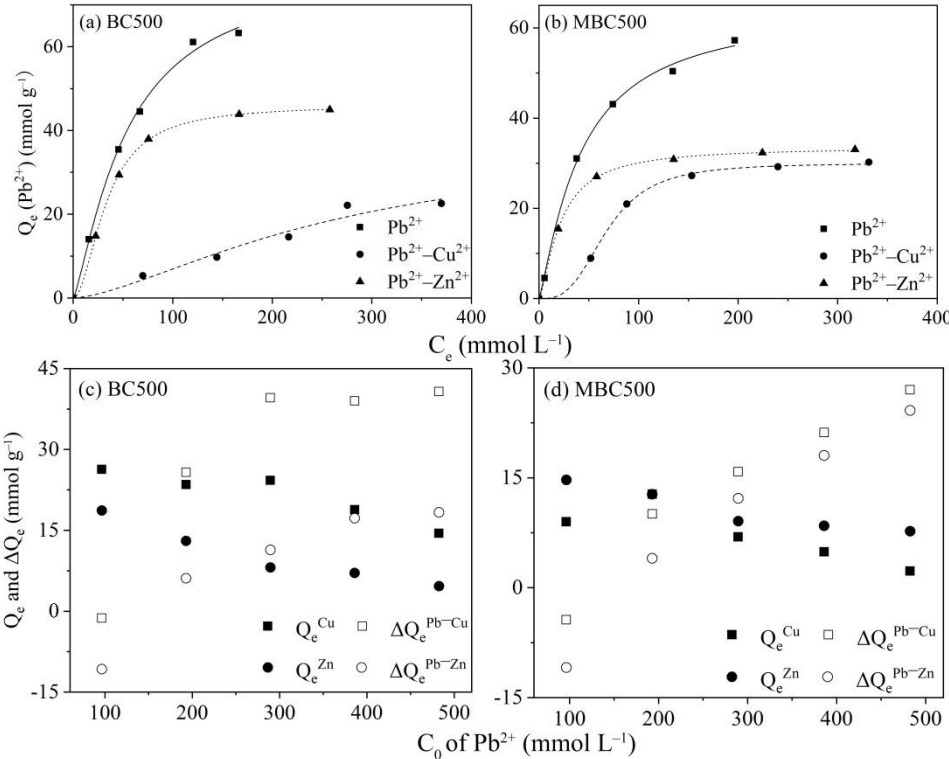

**Figure 2.** Effect of $Cu^{2+}/Zn^{2+}$ on adsorption capacity of $Pb^{2+}$ on BC500 (**a**,**c**) and MBC500 (**b**,**d**) ($Q_e^{Cu}$ and $Q_e^{Zn}$ represent the adsorption capacities of $Cu^{2+}$ and $Zn^{2+}$ in the binary metal system; $\Delta Q_e^{Pb-Cu}$ and $\Delta Q_e^{Pb-Zn}$ represent the $\Delta Q_e$ of $Pb^{2+}$ in $Pb^{2+}$–$Cu^{2+}$ and $Pb^{2+}$–$Zn^{2+}$ competitive systems).

### 3.2.3. Competitive Adsorption of $Zn^{2+}$ with $Cu^{2+}/Pb^{2+}$

As shown in Figure 3a,b, the competitive adsorption of $Zn^{2+}$ with $Cu^{2+}/Pb^{2+}$ changed the adsorption type and reduced the adsorption capacity of $Zn^{2+}$ on BC500 and MBC500. In the single-metal system, the adsorption isotherm of $Zn^{2+}$ could be determined as L-2 type, and there was a gentle plateau at the end of the curves. When there were other

competing ions (i.e., $Cu^{2+}$ and $Pb^{2+}$) in the system, the adsorption type changed into L-1 type. This result indicated that the adsorption capacity of $Zn^{2+}$ could be further increased with the changed concentration of metal ions in the binary metal system. According to Figure 3c,d, the $\Delta Q_e$ of $Zn^{2+}$ on BC500 and MBC500 was relatively stable and the adsorption capacities of $Cu^{2+}$ and $Pb^{2+}$ were slightly changed. Combined with the results in Table S8, the inhibition effect of $Zn^{2+}$ in the $Zn^{2+}$–$Cu^{2+}$ system was relatively weaker than that in the $Zn^{2+}$–$Pb^{2+}$ system; the reduction in adsorption capacity was 34.1% (BC500) and 53.2% (MBC500). The notable decreasing in adsorption of $Zn^{2+}$ in the presence of $Pb^{2+}$ could be illustrated according to the study of Zhao et al. [39]: the binding energy of $Pb^{2+}$ (139 eV) is far lower than that of $Zn^{2+}$ (1023 eV); thus, the oxygen-containing functional groups on BC500 and MBC500 have stronger binding ability to $Pb^{2+}$ than to $Zn^{2+}$. This result indicated that MBC500 is preferable for the removal of metal ions in a $Cu^{2+}$–$Zn^{2+}$ binary metal system.

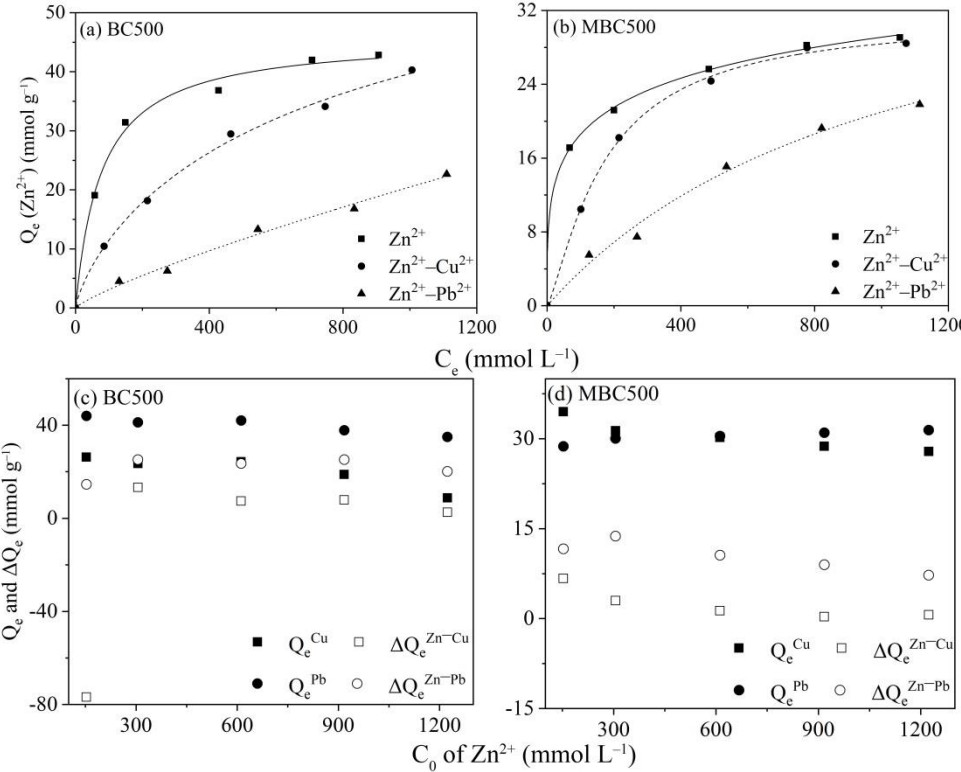

**Figure 3.** Effect of $Cu^{2+}$/$Pb^{2+}$ on adsorption capacity of $Zn^{2+}$ on BC500 (**a,c**) and MBC500 (**b,d**) ($Q_e^{Cu}$ and $Q_e^{Pb}$ represent the adsorption capacities of $Cu^{2+}$ and $Pb^{2+}$ in the binary metal system; $\Delta Q_e^{Zn-Cu}$ and $\Delta Q_e^{Zn-Pb}$ represent the $\Delta Q_e$ of $Zn^{2+}$ in $Zn^{2+}$–$Cu^{2+}$ and $Zn^{2+}$–$Pb^{2+}$ competitive systems).

### 3.3. Ternary Metal Adsorption onto Biochars

The results of ternary competitive adsorption are shown in Figure 4 and Table S9. The adsorption capacity of $Pb^{2+}$ onto BC500 and MBC500 in single and ternary metal systems was higher than that of $Cu^{2+}$ and $Zn^{2+}$. The single and ternary metal adsorption isotherms of $Cu^{2+}$ and $Zn^{2+}$ were classified as L-2 type. The adsorption isotherm of $Pb^{2+}$ changed from L-2 type (single-metal system) to L-1 type (ternary metal system), suggesting that the adsorption capacity of $Pb^{2+}$ could be further increased in ternary metal systems. As shown in Figure 4e,f, the $\Delta Q_e$ of $Cu^{2+}$ on MBC500 was higher than that on BC500, showing an increasing trend when the $C_0$ of the three metals was below 629.5 mmol $L^{-1}$.

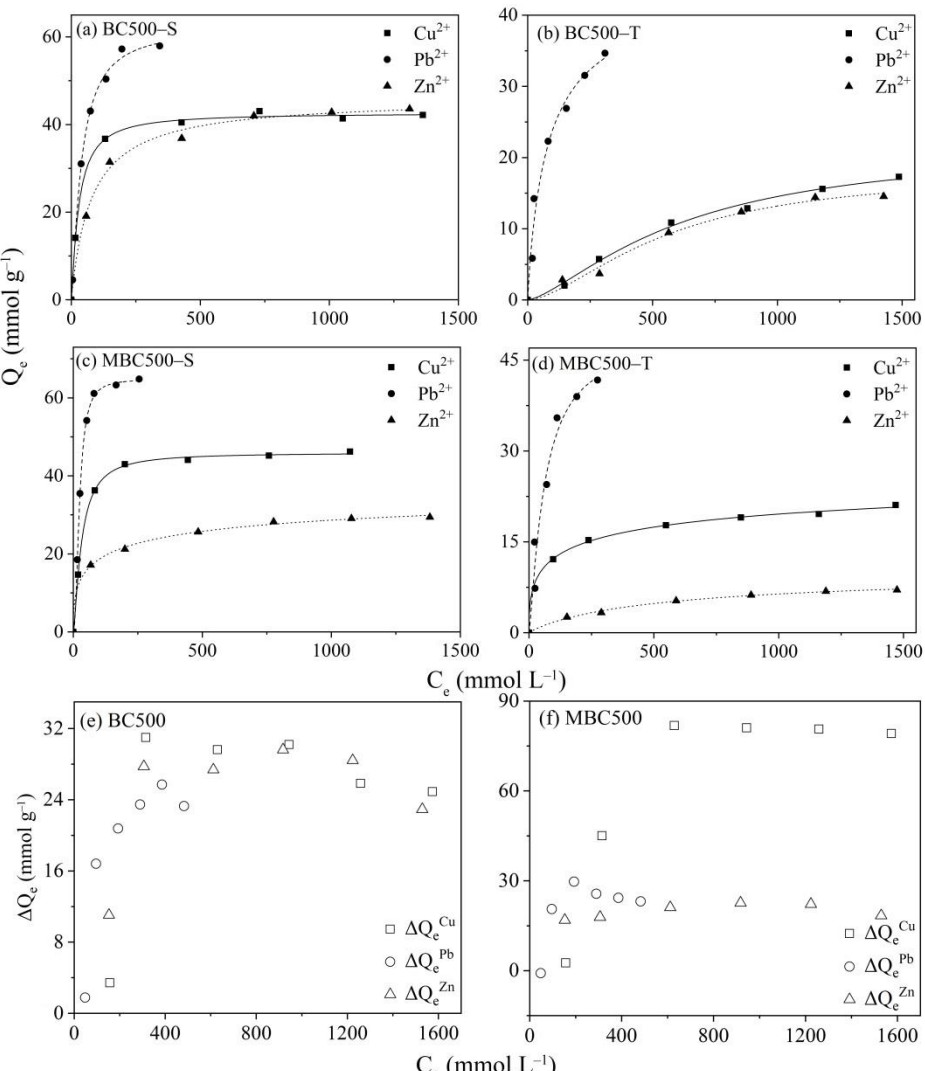

**Figure 4.** Competitive adsorption of $Cu^{2+}$, $Pb^{2+}$, and $Zn^{2+}$ (S: single metal adsorption (**a**,**c**); T: ternary metal adsorption (**b**,**d**); $Q_e^{Cu}$, $Q_e^{Pb}$ and $Q_e^{Zn}$ represent the adsorption capacity of $Cu^{2+}$, $Pb^{2+}$ and $Zn^{2+}$ in ternary-metal system, respectively; $\Delta Q_e^{Cu}$, $\Delta Q_e^{Pb}$, and $\Delta Q_e^{Zn}$ represent the $\Delta Q_e$ of $Cu^{2+}$, $Pb^{2+}$, and $Zn^{2+}$ in ternary metal systems, respectively (**e**,**f**)).

For BC500, the $Q_m$ of $Cu^{2+}$, $Pb^{2+}$, and $Zn^{2+}$ decreased from 42.66 to 21.58 mmol g$^{-1}$ (49.4%), 62.24 to 44.30 mmol g$^{-1}$ (28.8%), and 46.08 to 18.20 mmol g$^{-1}$ (60.5%), respectively. The adsorption of metal ions on MBC500 showed the same trends as that on BC500; the adsorption capacity of $Cu^{2+}$, $Pb^{2+}$, and $Zn^{2+}$ decreased from 45.96 to 31.53 mmol g$^{-1}$ (31.4%), 64.81 to 48.74 mmol g$^{-1}$ (24.8%), and 42.55 to 10.26 mmol g$^{-1}$ (75.9%), respectively. These results showed that the competitive adsorption order of the three heavy metal ions was $Pb^{2+} > Cu^{2+} > Zn^{2+}$. Qi and Aldrich [40] and Taha et al. [41] also reported similar orders of adsorption.

Indeed, the quantity of adsorbed ions on the biochar surface is related not only to the characteristics of adsorbents but also other factors, such as the hydration radius, electronegativity, and hydrolysis constant of metal ions. As mentioned above, the hydration radii of the three metal ions follow the order of $Pb^{2+} < Cu^{2+} < Zn^{2+}$, demonstrating that $Pb^{2+}$ could easily exchange ions on the surface of biochar. Surface electrostatic attraction is an important adsorption mechanism of metal ions on biochar, and metal ions with stronger electronegativity can be easily adsorbed on the surface of biochars. The order of electronegativity of the studied metal ions is $Pb^{2+}$ (2.33) > $Cu^{2+}$ (1.90) > $Zn^{2+}$ (1.65), indicating that $Pb^{2+}$ has a greater competitive advantage in adsorption [42]. The metal

adsorption affinity increases with the increasing hydrolysis constant of heavy metal ions, and the specific adsorption of adsorbents to ions decreases. The order of hydrolysis constants of the studied metal ions is $Pb^{2+}$ ($10^{-7.71}$) > $Cu^{2+}$ ($10^{-8}$) > $Zn^{2+}$ ($10^{-9}$), which is consistent with the metal adsorption capacity [39,43]. A previous study also demonstrated that metal ions with smaller ionic diameters have higher adsorption rates [44]. The ionic radii (Pauling) of $Cu^{2+}$, $Pb^{2+}$, and $Zn^{2+}$ are 0.72, 1.20, and 0.74 Å, respectively. In this study, $Pb^{2+}$, with the highest ionic radius, had the highest adsorption capacity, suggesting that the surface adsorption on adsorbents contributed to the larger adsorption capacity more than the microporous adsorption (which prefers smaller ions).

In this study, MBC500 had a slightly larger adsorption capacity for $Pb^{2+}$ than BC500 in the ternary metal system and was easy to separate from the solution, suggesting that MBC500 could be used to remove metal ions in an aqueous environment, especially for wastewater in which the major metal pollutant is $Pb^{2+}$. The maximum adsorption capacity was 48.74 mmol g$^{-1}$ (10.10 mg g$^{-1}$), being higher than some of the adsorbents in previous studies (Table S10).

### 3.4. Possible Mechanisms

### 3.4.1. Surface Characteristics

The $N_2$ adsorption–desorption isotherms of BCs and MBCs are presented in Figure 5a,b. All the isotherms presented a similar shape, which could be categorized as type IV according to the IUPAC (International Union of Pure and Applied Chemistry) classification. Generally, type IV isotherms are predominantly mesoporous, with the diameter range of 2–50 nm [45,46]. As shown in Table 2, the BET surface area, total pore volume, and average pore size of BCs and MBCs increased with the increasing pyrolysis temperature, which may be due to the organic matter and other volatile matter inside the pomelo peel having decomposed or volatilized more rapidly and thoroughly under high pyrolysis temperature. A large amount of energy was suddenly released, opening the pores inside the pomelo peel and increasing the porosity and surface area, which is beneficial to improve its adsorption capacity. Meanwhile, there was a smaller decrease in the surface area of the MBCs than that of the BCs, which is probably due to the surface-loaded $Fe_3O_4$ blocking some of the pores [47]. In addition, smaller micropore area and pore size were also observed for MBCs due to the collapse of micropores and blocking of iron, which is similar to the results of a previous study [48]. Combined with the analysis in kinetics, the average pore sizes of biochars were larger than the ion radii of three kinds of metals, so metal ions could enter the biochars. Thus, the internal diffusion contributed to the adsorption of heavy metals but is not the main adsorption mechanism.

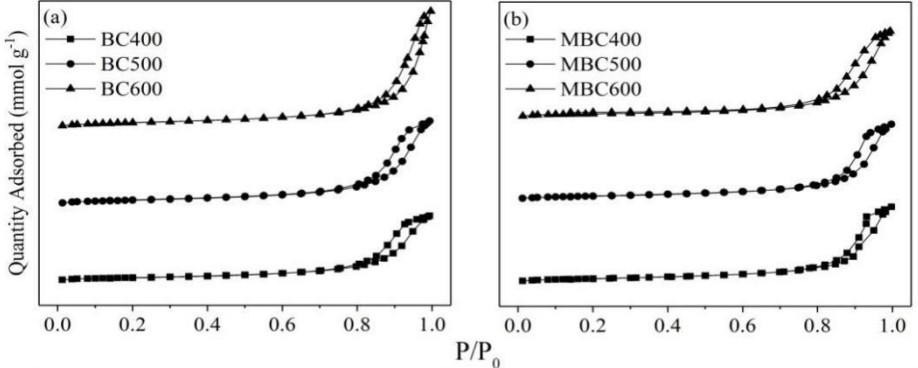

**Figure 5.** Adsorption–desorption isotherms of $N_2$ on BCs (**a**) and MBCs (**b**).

**Table 2.** Surface structural characteristics of BCs and MBCs. BET—Brunauer–Emmett–Teller.

| Biochar | BET Surface Area ($m^2\ g^{-1}$) | Micropore Area ($m^2\ g^{-1}$) | Total Pore Volume ($cm^3\ g^{-1}$) | Average Pore Size (nm) |
|---|---|---|---|---|
| BC400 | 21.65 | 4.22 | 0.11 | 19.98 |
| BC500 | 26.53 | 4.27 | 0.14 | 20.77 |
| BC600 | 32.13 | 5.34 | 0.17 | 23.08 |
| MBC400 | 19.97 | 4.92 | 0.10 | 18.82 |
| MBC500 | 22.44 | 2.33 | 0.12 | 19.77 |
| MBC600 | 26.48 | 1.76 | 0.14 | 21.97 |

### 3.4.2. FTIR Analysis

Figure S7 shows the FTIR spectra of pomelo peel biochars before and after adsorption of metal ions. The typical infrared absorption peaks of biochar were still maintained, indicating that the adsorption of heavy metal ions did not destroy the structure of biochar. Both BCs and MBCs had a broad and strong band centered at 3400 $cm^{-1}$, which is related to the stretching of the O–H bond of hydroxyl groups (–OH). After adsorption, the intensity of the –OH stretching band changed to a certain extent, suggesting that –OH contributed to heavy metal adsorption. The band at 2360 $cm^{-1}$ could be assigned to the O=C=O stretching vibrations, which shifted after the adsorption of metals. The strong peaks at 1650 $cm^{-1}$ presented the stretching vibrations of –COOH shifted after adsorption, indicating the involvement of carboxylate groups in the removal of $Cu^{2+}$, $Pb^{2+}$, and $Zn^{2+}$ [49,50].

The peaks at 1380 and 1470 $cm^{-1}$ indicated that groups of C=O existed in the biochars, which also changed the intensities after adsorption. The changes in shapes and intensities of the bands in the FTIR spectra implied that hydroxyl and carboxyl groups of biochars played important roles in the adsorption of metal ions. The abundant oxygen-containing functional groups and metal–oxygen bonds on the surface of biochar provide adsorption sites for heavy metal ions and promote the formation of metal ligand complexes, suggesting that ion exchange and complexation are possible mechanisms. The vibration peaks at 875, 854, and 774 $cm^{-1}$ are related to the C–H structures in olefins and aromatics [51]. For BC500 and MBC500, new low-frequency peak of carbonates could be observed at 946 $cm^{-1}$, and the peaks at 725 $cm^{-1}$ were amplified, which may be associated with the formation of $M(OH)_2$ and $M_3(CO_3)_2(OH)_2$ (M represents the studied metal element) [52]. In addition, MBCs had some different functional groups compared with BCs. The low-frequency peak observed at 583–632 $cm^{-1}$ on the MBCs was attributed to the vibration of Fe–O from $Fe_3O_4$, suggesting a successful load of $Fe_3O_4$ on the BCs [18].

### 3.4.3. XRD Analysis

In order to investigate the crystallographic structures of BCs and MBCs, XRD analyses before and after adsorption were carried out and the XRD patterns are shown in Figure S8. For BCs, the change trend of the three kinds of biochar was similar. Thus, the following analysis took BC500 as the object. Before adsorption, the main peaks at 2θ = 23.9° confirmed the presence of quartz. The peaks at 2θ = 28.7° and 2θ = 30.7° indicated the presence of $CaCO_3$ and $Ca_3(PO_4)_2$, respectively [53]. The XRD pattern of BC500 changed obviously after $Pb^{2+}$ adsorption. The peaks at 2θ = 20.0°, 27.2°, 34.1°, and 36° represented the formation of $Pb_3(CO_3)_2(OH)_2$, and the peaks at 2θ = 24.8°, 43.4°, 46.9°, and 49.0° represented the formation of $PbCO_3$ [52]. After the adsorption of $Cu^{2+}$, BC500 showed elevated weak peaks at 2θ = 29.0° and 36.2°, in the form of $Cu_2O$ [54]. New small peaks appearing at 2θ = 29.2° and 32.3° after the adsorption of $Zn^{2+}$ correspond to $Zn_3(PO_4)_2$ [55]. These findings confirmed that surface mineral participation was one of the possible mechanisms in the adsorption of the studied metals and the adsorption of $Cu^{2+}$, $Pb^{2+}$, and $Zn^{2+}$ on BC was a chemisorption process. For the MBCs, a large number of absorption peaks could be observed after magnetization. Strong and sharp peaks at 2θ = 35.6°, 43.3°, 53.6°, 57.3°, and 62.9° confirmed the presence of $Fe_3O_4$ on the MBCs, suggesting that iron was successfully loaded on the surface of the BCs [56]. Meanwhile, no peaks of impurities

were detected in the MBCs, indicating that the $Fe_3O_4$ was highly crystalline with high purity [57]. Comparing the XRD patterns before and after adsorption, it can be seen that that there was no obvious change in the XRD characteristic peaks of MBCs, suggesting that the crystal structure of $Fe_3O_4$ remained stable and the magnetic properties and structural stability of the MBCs could be maintained.

Based on the results of FTIR and XRD, the possible adsorption mechanisms of $Cu^{2+}$, $Pb^{2+}$, and $Zn^{2+}$ are as follows: (1) ion exchange; (2) complexation between metal ions and the oxygenated functional groups on biochar; (3) surface mineral precipitation of metal ions.

## 4. Conclusions

BCs and MBCs of pomelo peel were prepared. The adsorption performances of heavy metal ions by BCs and MBCs under single and multi-metal systems were evaluated. Increasing pyrolysis temperature could change the physico-chemical properties of pomelo peel biochars, and iron was successfully loaded on the surface of biochars. The adsorption of $Cu^{2+}$, $Pb^{2+}$, and $Zn^{2+}$ on BCs and MBCs was a chemisorption and monolayer adsorption process. The adsorption of metal ions on the biochars was not only controlled by the intraparticle diffusion. The biochar obtained at 500 °C had a good adsorption capacity for the tested metal ions, and the adsorption capacity ranged as $Pb^{2+}$ and $Cu^{2+} > Zn^{2+}$. The adsorption of metal ions on the biochar was a spontaneous and feasible endothermic process. The adsorption isotherms of $Pb^{2+}$ and $Zn^{2+}$ changed in binary metal systems, and only that of $Pb^{2+}$ changed in ternary metal systems. The competitive adsorption order of the three metal ions was $Pb^{2+} > Cu^{2+} > Zn^{2+}$. Comprehensive results showed that MBC500 could be applied to removal heavy metal ions in wastewater in which the major metal pollutant is $Pb^{2+}$ and the maximum adsorption capacity could be up to 48.74 mmol g$^{-1}$. The possible mechanisms were ion exchange, complexation, and surface mineral precipitation of metal ions. Furthermore, this study also provided technical support for the utilization of pomelo peel and the engineering application of biochar.

**Supplementary Materials:** The following are available online at https://www.mdpi.com/2073-4441/13/6/868/s1. Figure S1: Preparation of biochars (BC: biochar, MBC: magnetic biochar), Figure S2: Adsorption kinetics of $Cu^{2+}$ (**a**,**b**), $Pb^{2+}$ (**c**,**d**) and $Zn^{2+}$ (**e**,**f**) on BCs and MBCs, Figure S3: Intraparticle diffusion plots of adsorption for $Cu^{2+}$ (**a**,**b**), $Pb^{2+}$ (**c**,**d**) and $Zn^{2+}$ (**e**,**f**) on BCs and MBCs, Figure S4: Adsorption thermodynamics of $Cu^{2+}$ on BCs (**a**,**c**,**e**) and MBCs (**b**,**d**,**f**), Figure S5: Adsorption thermodynamics of $Pb^{2+}$ on BCs (**a**,**c**,**e**) and MBCs (**b**,**d**,**f**), Figure S6: Adsorption thermodynamics of $Zn^{2+}$ on BCs (**a**,**c**,**e**) and MBCs (**b**,**d**,**f**), Figure S7: FTIR spectra of BCs (**a**,**c**,**e**) and MBCs (**b**,**d**,**f**) before and after adsorption of $Cu^{2+}$, $Pb^{2+}$ and $Zn^{2+}$, Figure S8: XRD patterns of BCs (**a**,**c**,**e**) and MBCs (**b**,**d**,**f**) before and after adsorption of $Cu^{2+}$, $Pb^{2+}$ and $Zn^{2+}$, Table S1: Kinetic parameters of the pseudo-second-order model for metal ions adsorption on BCs and MBCs, Table S2: Kinetic parameters of the intraparticle diffusion model for metal ions adsorption on BCs and MBCs, Table S3: Fitting results of adsorption thermodynamics of $Cu^{2+}$ on BCs and MBCs, Table S4: Fitting results of adsorption thermodynamics of $Pb^{2+}$ on BCs and MBCs, Table S5: Fitting results of adsorption thermodynamics of $Zn^{2+}$ on BCs and MBCs, Table S6: Fitting results of Langmuir model for competitive adsorption of $Cu^{2+}$ with $Pb^{2+}/Zn^{2+}$, Table S7: Fitting results of Langmuir model for competitive adsorption of $Pb^{2+}$ with $Cu^{2+}/Zn^{2+}$, Table S8: Fitting results of Langmuir model for competitive adsorption of $Zn^{2+}$ with $Cu^{2+}/Pb^{2+}$, Table S9: Fitting results of Langmuir model for competitive adsorption of $Cu^{2+}$, $Pb^{2+}$ and $Zn^{2+}$, Table S10: Maximum adsorption capacities of some adsorbents for $Pb^{2+}$ in previous studies.

**Author Contributions:** Conceptualization, L.W.; methodology, S.D.; validation, L.W. and S.D.; formal analysis, L.W.; investigation, S.D.; resources, L.W. and X.L.; writing—original draft preparation, Q.W.; writing—review and editing, L.W. and X.L. All authors have read and agreed to the published version of the manuscript.

**Funding:** This research was funded by the National Natural Science Foundation of China (41877516 and 41703093) and the Natural Science Foundation of Shaanxi Province (2019JM-101).

**Institutional Review Board Statement:** Not applicable.

**Informed Consent Statement:** Not applicable.

**Data Availability Statement:** Data sharing not applicable.

**Conflicts of Interest:** The authors declare no conflict of interest.

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
