# Peer review of "Single and Competitive Adsorption Behaviors of Cu2+, Pb2+ and Zn2+ on the Biochar and Magnetic Biochar of Pomelo Peel in Aqueous Solution"

_water, doi:10.3390/w13060868_

Round 1

Reviewer 1 Report

The article presented single and competitive adsorption behaviors of Cu2+, Pb2+ and  Zn2+ on the biochar and magnetic biochar of pomelo peel in aqueous solution. Research on biocarbon as an environment-friendly material, biochar has been used to remove heavy metals from wastewater is very interesting and important due to the development of the ecological trend in environmental engineering. The materials and methods, the results were described and presented very well and precisely. However, the authors did not do without a few editing errors. Below I present my comments and questions to the author.

Introduction

In the introduction, we should mention the influence of magnetic fields on water treatment (article for review and quotation - DOI: 10.3390 / w11081584) and the removal of heavy metal ions in the magnetic field assisted filtration process (article for literature review - DOI: 10.5004 / dwt. 2018.22551 )

Materials and methods

In the section "Preparation of biochars" a diagram would be useful, which would illustrate step by step the stages of producing both sorbents

The research methods should be described in more detail in the subsection "Characterization of biochars"

Results and Discussion

In the subsection "FTIR analysis" and in the subsection "\ XRD analysis ”should be a more extensive discussion.

Author Response

Reviewer#1

The article presented single and competitive adsorption behaviors of Cu2+, Pb2+ and  Zn2+ on the biochar and magnetic biochar of pomelo peel in aqueous solution. Research on biocarbon as an environment-friendly material, biochar has been used to remove heavy metals from wastewater is very interesting and important due to the development of the ecological trend in environmental engineering. The materials and methods, the results were described and presented very well and precisely. However, the authors did not do without a few editing errors. Below I present my comments and questions to the author.

Introduction

Comment 1: In the introduction, we should mention the influence of magnetic fields on water treatment (article for review and quotation - DOI: 10.3390 / w11081584) and the removal of heavy metal ions in the magnetic field assisted filtration process (article for literature review - DOI: 10.5004 / dwt. 2018.22551 )

Response: Than you very much for your suggestions. The influence of magnetic fields on water treatment and the removal of heavy metal ions in the magnetic field assisted filtration process have been added in line 55-60 in the second paragraph of introduction. Meanwhile, two references (i.e., 20 and 21) are also given. Thanks.

Materials and methods

Comment 2: In the section "Preparation of biochars" a diagram would be useful, which would illustrate step by step the stages of producing both sorbents

Response: The process map of biochars preparation has been provided. However, there have been 5 figures in the manuscript and 7 figures in supplemental materials. Therefore, it is also placed in supplemental materials as figure S1. Thank you very much.

Comment 3: The research methods should be described in more detail in the subsection "Characterization of biochars"

Response: The characterization of biochars been improved in line 105-109 and 110- 115. Thanks for your suggestions.

Results and Discussion

Comment 4: In the subsection "FTIR analysis" and in the subsection "\ XRD analysis ”should be a more extensive discussion.

Response: The results of FTIR and XRD analyses have been revised in line 469-471 and 481-485 as well as 505-507. Thank you very much.

In addition, we also carefully check the whole manuscript.

Reviewer 2 Report

I prepared the manuscript review Single and competitive adsorption behaviors of Cu2+, Pb2+ and 2 Zn2+ on the biochar and magnetic biochar of pomelo peel in 3 aqueous solution written by Ioannis Qianlan Wu, Shuzhen Dong, Lijun Wang, and Xiaoyun Li.

The study is clear and adequately described. The approach is some what conventional, but the results will be interest to some readers, which are interested in adsorption on halloysite adsorbents. In my opinion this manuscript is acceptable for publication.

Author Response

Thank you very much for your positive comments and affirmations on our work.

Reviewer 3 Report

This paper studies the adsorption of Pb(II), Cu(II) and Zn(II) in single-, binary- and ternary-metal systems using a magnetic and non-magnetic pomelo peel biochar. This manuscript represent a very nice piece of work. It is very well structured and discussed. A number of characterization techniques were used and they are also well described. The adsorption kinetic and thermodynamic processes were also thoroughly described. The authors also compared some results with the literature. In summary, I recommend this paper for publication in Water. Some minor comments:

  • In the preparation procedure it is not mentioned the carbonization stage of the magnetic biochars at different temperatures. Were they separated by the magnetic field before or after the carbonization stage? Were they washed with distilled water before or after the carbonization stage?
  • Which is the amount of Fe retained in the magnetic biochars?. Can it been lixiviated during the adsorption experiments?
  • Line 96: The Barret-Joyner-Halender (BJH) method was applied to calculate the pore size distribution. Was the BJH method applied to the adsorption or desorption isotherm?. It should be clarify.
  • The reader would have expected to read the characterization of the fresh biochar before the adsorption experiments. As for example, the textural characteristics of the materials prepared. The authors present the characterization results after the adsorption experiments.
  • During the XRD the authors detect the presence of CaCO3 and Ca3(PO4)2 in the biochar before adsorption. How it can affect to the adsorption experiments?.

Author Response

Reviewer#3

This paper studies the adsorption of Pb(II), Cu(II) and Zn(II) in single-, binary- and ternary-metal systems using a magnetic and non-magnetic pomelo peel biochar. This manuscript represent a very nice piece of work. It is very well structured and discussed. A number of characterization techniques were used and they are also well described. The adsorption kinetic and thermodynamic processes were also thoroughly described. The authors also compared some results with the literature. In summary, I recommend this paper for publication in Water.

Some minor comments:

Comment 1: In the preparation procedure it is not mentioned the carbonization stage of the magnetic biochars at different temperatures. Were they separated by the magnetic field before or after the carbonization stage? Were they washed with distilled water before or after the carbonization stage?

Response: The preparation process has been revised. As described in the biochars preparation of 2.2, the magnetic biochars were prepared by chemical co-precipitation method with the prepared non-magnetic pomelo peel biochars at different temperatures as materials. Thus, the preparation process of magnetic biochars has no carbonization stage. The prepared magnetic biochars were separated from solution by magnetic field. The prepared magnetic and non-magnetic biochars were all washed to neutral with distilled water. Thank you.

Comment 2: Which is the amount of Fe retained in the magnetic biochars?. Can it been lixiviated during the adsorption experiments?

Response: In fact, the amount of Fe retained in the magnetic biochars and the lixiviation of Fe from magnetic biochars were not determined in this study. They will be studied in our futural investigation into the reuse of magnetic biochars. Thank you very much for your kind reminder.

Comment 3: Line 96: The Barret-Joyner-Halender (BJH) method was applied to calculate the pore size distribution. Was the BJH method applied to the adsorption or desorption isotherm?. It should be clarify.

Response: It has been clarified in line 107-109. Thank you very much.

Comment 4: The reader would have expected to read the characterization of the fresh biochar before the adsorption experiments. As for example, the textural characteristics of the materials prepared. The authors present the characterization results after the adsorption experiments.

Response: Thank you for pointing this out. In the experimental design, the biochars before and after adsorption were characterized by FTIR, XRD and SEM analysis. Because the analysis of magnetic samples was not allowed for SEM, the SEM analysis results of magnetic biochars before and after adsorption could not be obtained, and the results cannot be compared with the non-magnetic biochar. Therefore, only the FTIR and XRD results before and after adsorption were provided in this paper and supplementary materials.

Comment 5: During the XRD the authors detect the presence of CaCO3 and Ca3(PO4)2 in the biochar before adsorption. How it can affect to the adsorption experiments?.

Response: As mentioned in 3.4.3., for XRD analysis (takes BC500 as the object), CaCO3 and Ca3(PO4)2 in BC500 were detected before adsorption; Pb3(CO3)2(OH)2, PbCO3 and Zn3(PO4)2 were detected after adsorption; which confirmed that the surface mineral participation was one of the possible mechanisms in the adsorption of the studied metal ions. Thank you very much.

Reviewer 4 Report

he manuscript presents a comprehensive batch study on competitive adsorption of three heavy metals on biochar and magnetic biochar. The researchers have conducted broad studies on the adsorption kinetics and equilibrium studies on mono-, di, and tri-ion adsorption studies that provide some insight. The quality of this paper is good and I support the publication of this paper, after some revision.

Here are some questions that the author ought to consider while reviewing the manuscript:

  1. The concentration of the metal ion used for this study ranges from 10 mg/L to 100 mg/L, which is a lot higher than what is commonly found in contaminated surface waters. How are the results of this study transferable to more diluted concentrations?
  2. The range of the different metal concentrations was different. For example, Cu2+ ranged 10 – 80 mg/L, Pb2+ (50 to 200 mg/L) and Zn2+ (5 – 50 mg/L) were tested. What was the rationale for the selection of different concentrations of the individual metal ions, while the total concentrations are high and rarely found in surface waters?
  3. The role of the anions in the systems studied is not discussed. How does the aqueous solution equilibrium of the three ions affect their adsorption behavior?
  4. The difference in hydration radii of the three ions (Line 313) may not explain the difference in adsorption capacity. The average pore size for the three types of adsorbent is about 20 nm which is about 50 times than hydration radii of the ions. Providing more explanation could help.
  5. The surface chemistry of the biochars prepared at different temperatures is similar. However, no data is provided on the charge (zeta potential) of the surfaces. What is the role of surface charge on adsorption capacities?
  6. Studying competitive adsorption of co-existing metal ions is important. However, that study does not include natural organic matter which is ubiquitous in the natural environment.

Author Response

Reviewer#4

The manuscript presents a comprehensive batch study on competitive adsorption of three heavy metals on biochar and magnetic biochar. The researchers have conducted broad studies on the adsorption kinetics and equilibrium studies on mono-, di, and tri-ion adsorption studies that provide some insight. The quality of this paper is good and I support the publication of this paper, after some revision.

Here are some questions that the author ought to consider while reviewing the manuscript:

Comment 1: The concentration of the metal ion used for this study ranges from 10 mg/L to 100 mg/L, which is a lot higher than what is commonly found in contaminated surface waters. How are the results of this study transferable to more diluted concentrations?

Response: In this study, in order to obtain the kinetics curves and adsorption isotherm, the pre-experiments of adsorption were carried out for different heavy metal ions. Then, the concentration (adsorption kinetics) and concentration range (adsorption isotherm) of different heavy metal ions were set up based on the pre-experimental results, which were higher than those in commonly contaminated surface water. However, the results of this study may still provide some theoretical basis for some industrial water treatment, such as metallurgical industry and papermaking wastewater. The biochar in this study may not have the best adsorption efficiency at low heavy metals concentrations, but it also has a better removal effect in a specific concentration range. In addition, magnetic biochar is easy to separate, so it could be suitable for some wastewater treatment. Thank you very much.

Comment 2: The range of the different metal concentrations was different. For example, Cu2+ ranged 10 – 80 mg/L, Pb2+ (50 to 200 mg/L) and Zn2+ (5 – 50 mg/L) were tested. What was the rationale for the selection of different concentrations of the individual metal ions, while the total concentrations are high and rarely found in surface waters?

Response: See the response to comment 1.

Comment 3: The role of the anions in the systems studied is not discussed. How does the aqueous solution equilibrium of the three ions affect their adsorption behavior?

Response: This study mainly focused on the adsorption kinetics and adsorption isotherm of single-metal ion on the (magnetic) biochars as well the competitive adsorption of binary- and ternary-metal ions on the (magnetic) biochars. For the influence factors (e.g., solution pH, natural organic matter NOM, anions and cations) of adsorption and the reuse of biochars, they will be conducted in futural investigation. Thank you very much for your suggestions.

Comment 4: The difference in hydration radii of the three ions (Line 313) may not explain the difference in adsorption capacity. The average pore size for the three types of adsorbent is about 20 nm which is about 50 times than hydration radii of the ions. Providing more explanation could help.

Response: The explanation has been updated in line 327-332. Thanks.

Comment 5: The surface chemistry of the biochars prepared at different temperatures is similar. However, no data is provided on the charge (zeta potential) of the surfaces. What is the role of surface charge on adsorption capacities?

Response: At present, the zeta potential of biochar surfaces could not be measured in our laboratory. Thank you very much.

Comment 6: Studying competitive adsorption of co-existing metal ions is important. However, that study does not include natural organic matter which is ubiquitous in the natural environment.

Response: Same to the response to comment 3. Thanks.

Round 2

Reviewer 1 Report

All my comments were properly taken into account. The article is suitable for publication in its current form.